# Mental disorders and intimate partner violence perpetrated by men towards women: A Swedish population-based longitudinal study

Rongqin Yu[1], Alejo J. Nevado-Holgado[1], Yasmina Molero[1], Brian M. D'Onofrio[2,3], Henrik Larsson[3,4], Louise M. Howard[5], Seena Fazel[1] *

1 Department of Psychiatry, University of Oxford, Oxford, United Kingdom, 2 Department of Psychological and Brain Sciences, Indiana University, Bloomington, Indiana, United States of America, 3 Department of Medical Epidemiology and Biostatistics, Karolinska Institute, Stockholm, Sweden, 4 School of Medical Sciences, Örebro University, Örebro, Sweden, 5 Department of Women & Children's Health, King's College London, London, United Kingdom

* seena.fazel@psych.ox.ac.uk

**Data Availability Statement:** The Public Access to Information and Secrecy Act in Sweden prohibits us from making individual level data publicly

## Abstract

### Background

Intimate partner violence (IPV) against women is associated with a wide range of adverse outcomes. Although mental disorders have been linked to an increased risk of perpetrating IPV against women, the direction and magnitude of the association remain uncertain. In a longitudinal design, we examined the association between mental disorders and IPV perpetrated by men towards women in a population-based sample and used sibling comparisons to control for factors shared by siblings, such as genetic and early family environmental factors.

### Methods and findings

Using Swedish nationwide registries, we identified men from 9 diagnostic groups over 1998–2013, with sample sizes ranging from 9,529 with autism to 88,182 with depressive disorder. We matched individuals by age and sex to general population controls (ranging from 186,017 to 1,719,318 controls), and calculated the hazard ratios of IPV against women. We also estimated the hazard ratios of IPV against women in unaffected full siblings (ranging from 4,818 to 37,885 individuals) compared with the population controls. Afterwards, we compared the hazard ratios for individuals with psychiatric diagnoses with those for siblings using the ratio of hazard ratios (RHR). In sensitivity analyses, we examined the contribution of previous IPV against women and common psychiatric comorbidities, substance use disorders and personality disorders. The average follow-up time across diagnoses ranged from 3.4 to 4.8 years. In comparison to general population controls, all psychiatric diagnoses studied except autism were associated with an increased risk of IPV against women in men, with hazard ratios ranging from 1.5 (95% CI 1.3–1.7) to 7.7 (7.2–8.3) (p-values < 0.001). In sibling analyses, we found that men with depressive disorder, anxiety disorder, alcohol use

available. Researchers who are interested in replicating our work can apply for individual level data from: Statistics Sweden (mikrodata@scb.se) for data from The Total Population Register (https://www.scb.se/vara-tjanster/bestalla-mikrodata/vilka-mikrodata-finns/individregister/registret-over-totalbefolkningen-rtb/), The Multi-Generation Register (https://www.scb.se/vara-tjanster/bestalla-mikrodata/vilka-mikrodata-finns/individregister/flergenerationsregistret/), The Longitudinal Integrated Database for Health Insurance and Labour Market Studies (https://www.scb.se/en/services/guidance-for-researchers-and-universities/vilka-mikrodata-finns/longitudinella-register/longitudinal-integrated-database-for-health-insurance-and-labour-market-studies-lisa/), The National Board of Health and Welfare (registerservice@socialstyrelsen.se) for data from The Patient Register (https://www.socialstyrelsen.se/patientregistret), and The Swedish National Council for Crime Prevention (statistik@bra.se) for data from The National Crime Register (https://www.bra.se/statistik/kriminalstatistik/specialbestallningar.html).

**Funding:** SF is funded by a Wellcome Trust Senior Research Fellowship (grant number 202836/Z/16/Z). The funder of the study had no role in study design, data collection, data analysis, data interpretation, or writing of the report.

**Competing interests:** I have read the journal's policy and the authors of this manuscript have the following competing interests: AJNH is funded by Johnson and Johnson and received funds from Ono Pharma in 2018. HL has served as a speaker for Evolan Pharma and Shire and has received research grants from Shire; all outside the submitted work. LMH is the lead investigator on a UK Research and Innovation funded Mental Health Network on Violence, Abuse and Mental Health.

**Abbreviations:** ADHD, attention deficit hyperactivity disorder; IPV, intimate partner violence; RHR, ratio of hazard ratios.

disorder, drug use disorder, attention deficit hyperactivity disorder, and personality disorders had a higher risk of IPV against women than their unaffected siblings, with RHR values ranging from 1.7 (1.3–2.1) to 4.4 (3.7–5.2) ($p$-values < 0.001). Sensitivity analyses showed higher risk of IPV against women in men when comorbid substance use disorders and personality disorders were present, compared to risk when these comorbidities were absent. In addition, increased IPV risk was also found in those without previous IPV against women. The absolute rates of IPV against women ranged from 0.1% to 2.1% across diagnoses over 3.4 to 4.8 years. Individuals with alcohol use disorders (1.7%, 1,406/82,731) and drug use disorders (2.1%, 1,216/57,901) had the highest rates. Our analyses were restricted to IPV leading to arrest, suggesting that the applicability of our results may be limited to more severe forms of IPV perpetration.

## Conclusions

Our results indicate that most of the studied mental disorders are associated with an increased risk of perpetrating IPV towards women, and that substance use disorders, as principal or comorbid diagnoses, have the highest absolute and relative risks. The findings support the development of IPV risk identification and prevention services among men with substance use disorders as an approach to reduce the prevalence of IPV.

## Author summary

### Why was this study done?

- Intimate partner violence (IPV) perpetrated by men towards women is a global public health challenge and is associated with a range of poor outcomes in victims.

- One of the risk factors for IPV perpetration is mental disorders, but the nature and strength of the links with these disorders is uncertain, as previous studies typically measured the presence of mental disorders and perpetration of IPV at the same time, were based on small numbers, relied on self-report measures of IPV, and did not fully consider confounding factors including genetic and early family environmental factors.

### What did the researchers do and find?

- We identified men with common psychiatric disorders from a population-based sample, and compared their risk of IPV against women with that of age- and sex-matched general population controls, and also with that of their unaffected siblings to account for possible confounding familial factors.

- The absolute rate of IPV against women ranged from 0.1% for men with autism to 2.1% for men with drug use disorders. Most of the studied mental disorders were associated with a higher risk of IPV against women. The risk increase was 2 to 8 times compared with the general population and 2- to 4-fold compared with unaffected siblings.

- The highest absolute rates and relative risks for IPV perpetration were found in men with substance use disorders, and substance use comorbidity was associated with an elevated risk of IPV in other mental disorders.

## What do these findings mean?

- We found that several common mental disorders are associated with increased risk of IPV against women, and the risk is further elevated when there is a comorbidity with substance use disorders.

- Prevention and intervention programs should consider prioritizing assessment and treatment of IPV perpetration among individuals with psychiatric disorders, particularly those with alcohol and drug use disorders.

- Although the relative risk of IPV against women was higher in men with mental disorders, absolute rates of IPV were low. To reduce IPV against women, other modifiable risk factors need to be addressed.

## Introduction

Intimate partner violence (IPV) against women is a major public health problem. It is the most common form of violence experienced by women and includes physical, sexual, and emotional abuse and controlling behaviors by an intimate partner [1]. Estimates of the prevalence of IPV against women vary widely depending on the definitions. Worldwide, around 30% of women have experienced physical or sexual violence by their current or previous intimate partner [2]. IPV is associated with a wide range of serious health consequences in victims, such as physical injuries, pregnancy termination, sexually transmitted diseases, post-traumatic stress disorder, depression, and suicidality [3–7]. In addition, children exposed to IPV often develop a wide range of physical health, mental health, and social adjustment problems [8].

One potential risk factor for perpetrating IPV against women is mental illness, and etiological links may differ between different disorders. Common deficits associated with mental disorders, such as poor interpersonal skills and emotional dysregulation [9,10], and specific core symptoms of certain disorders—such as impulsivity manifested in individuals with attention deficit hyperactivity disorder (ADHD) and substance use disorders [11,12] and hostility exhibited in some people with mood disorders and antisocial personality disorder [13–15]—have been linked to IPV against women [15,16]. Preliminary evidence suggests that individuals with mental illness have increased risk of perpetrating IPV against women [17,18]. Systematic reviews have reported an increased risk of IPV perpetration among individuals with a range of mental disorders including depression, anxiety disorders, panic disorders, substance use disorders, and personality disorders, particularly antisocial personality disorder and borderline personality disorder [19–22]. However, the majority of existing empirical studies have been conducted with small sample sizes, have been based on selected samples, have relied on self-report of risk factors and outcomes, and, most importantly, have lacked adequate adjustment of confounders such as familial factors. There is considerable imprecision in previous work, partly due to different methodologies and samples. For instance, the hazard ratio of physical

violence against a partner by men has ranged from 1.7 to 5.5 for depression and from 0.8 to 9.1 for anxiety disorder [20].

Most of the evidence to date suggests some associations between mental disorders and IPV against women, but these associations might reflect underlying confounders or reverse direction of effects. Thus, the magnitude and direction of the links between mental disorders and IPV perpetration need clarification. In addition, the evidence for some disorders is very limited, particularly for specific psychiatric disorders, including schizophrenia-spectrum disorders, and developmental disorders such as ADHD and autism. For instance, autism, which is characterized by abnormal development of communication and social interaction [23], has been proposed to be associated with IPV as a result of impaired theory of mind, poor emotional regulation, and problems with moral reasoning [24]. In addition, health services research has suggested that it is especially difficult to help men with autism due to their inability to appreciate their partner's perspective, especially when arguments occur. Furthermore, the role of common psychiatric comorbidities, such as substance use disorders and personality disorders, is unclear.

Clarifying these associations can assist in developing more effective prevention and intervention programs [25]. To date, many such programs targeting perpetrators of IPV typically have limited effectiveness [26], and this may be partly related to the lack of modifiable factors in these programs.

Therefore, the aim of this study is to address these uncertainties in the association between mental disorders and men's IPV against women. To this end, we investigated risk of IPV against women among men with mental disorders in a population-based longitudinal cohort. As familial factors, such as genetic predisposition and shared childhood adversity, are associated with both mental disorders and IPV perpetration [27], we conducted sibling comparisons to control for familial confounders, and we also conducted a range of sensitivity analyses to identify potential moderators. To our knowledge, this is the largest epidemiological study of IPV perpetrators to date and the first to use sibling comparisons.

## Methods

### Study population and design

We used the unique 10-digit personal identification number [28] assigned to each Swedish resident to link several national registers in Sweden: the National Patient Register [29], the National Crime Register, the Multi-Generation Register (Statistics Sweden) [29], and the Longitudinal Integration Database for Health Insurance and Labour Market Studies.

We selected a cohort of individuals born between 1 January 1958 and 31 December 1998, who were followed from 1 January 1998 to the end of follow-up on 31 December 2013. We started our follow-up on 1 January 1998 as records of IPV perpetration in the National Crime Register started at that time. In this study, we focused only on IPV perpetrated by men towards women, which is recorded as a separate category of crime in the crime register. We could not examine IPV perpetrated by women towards men as current data in the Swedish registers do not separate this type of crime from general domestic violence in women. Arrests for male-to-female IPV were retrieved from the National Crime Register using a distinct crime code (0412), which is a unique advantage over crime data from many other countries where such a code is absent. The project was approved by the Regional Ethics Review Board in Stockholm, Sweden (2013/5:8), which waived the need for informed consent as anonymized register-based data were used.

## Mental disorder classifications

We studied 9 psychiatric disorders diagnosed in either an inpatient or outpatient setting between 1998 and 2013: schizophrenia-spectrum disorders, bipolar disorder, depressive disorder, anxiety disorder, alcohol use disorder, drug use disorder, ADHD, autism, and personality disorders. These disorders were classified according to ICD-10, using the following codes: schizophrenia-spectrum disorders (F20–F29), bipolar disorder (F30, F31), depressive disorder (F32–F39), anxiety disorder (F40–F42, F44–F45, F48), alcohol use disorder (F10 except x.5), drug use disorder (F11–F19 except x.5), ADHD (F90), autism (F84.0, F84.1), and personality disorders (F60). We adopted a hierarchical approach to the following diagnoses: schizophrenia-spectrum disorders, bipolar disorder, depressive disorder, and anxiety disorder, as research has shown that some diagnoses change over time (to a more stable one). For instance, depression and anxiety can be precursors of schizophrenia-spectrum disorders [30,31]. Thus, individuals with any diagnosis of bipolar disorder but not schizophrenia-spectrum disorder were regarded as having bipolar disorder. Individuals with any diagnosis of depressive disorder but neither schizophrenia-spectrum disorder nor bipolar disorder were regarded as having depressive disorder. Individuals with any diagnosis of anxiety disorder but without schizophrenia-spectrum disorder, bipolar disorder, and depressive disorder were considered as having anxiety disorder. This approach is expected to increase the validity of the above-mentioned mental disorders but could risk potentially underestimating some comorbidities. Therefore, we included comorbidities for common disorders including substance use disorders and personality disorders in the sensitivity analyses (see the statistical analyses section below). For diagnoses of alcohol use disorder, drug use disorder, ADHD, autism, and personality disorders, no hierarchical approach was assigned. Therefore, these disorders included both primary and secondary diagnoses. Diagnoses identified before arrest for IPV during 1998–2013 were defined as the exposure in this study. Swedish register-based psychiatric diagnoses generally have moderate to high concordance rates with clinical diagnoses [29].

## Outcome measure

Data for arrests for IPV between 1 January 1998 and 31 December 2013 were retrieved for all individuals in the cohort from the National Crime Register. This register includes crime data for all individuals aged 15 years (the age of criminal responsibility) and older. As a minority of IPV arrests result in conviction [32], we used first IPV arrest after diagnosis as our primary outcome. IPV against women is defined as threats, violence, and sexual assaults where the victim is a woman and the current or ex-partner is the offender (criminal code: 0412). In a sensitivity analysis, we included general domestic violence as the outcome, defined as violence against a person that the offender has or has had a close relationship to, including partners, children, parents, and siblings of the offender (criminal codes: 0411, 0412, 0422, 0423, 0424, 0425, 0440, 0441, 0442, and 0443). The National Crime Register has 99% coverage of the population [33].

## Sociodemographic covariates

We collected information on the following covariates: family disposable income, single status, and immigrant status. Family disposable income at the year of recorded diagnosis was used as a proxy for income and was treated as a dichotomous variable (i.e., lowest tertile versus middle and top tertiles). For the 2 developmental disorders (ADHD and autism), as the patients were relatively young, with nearly half lacking income data, we used the disposable income data of their parents. Single status was defined according to the year of diagnosis, and referred to

individuals who were unmarried, divorced, or widowed. Immigrant status was defined as being born outside of Sweden.

## Statistical analyses

We designed the analytic strategy when the study was conceived including the exposures (main psychiatric diagnoses), outcome (arrests for IPV), and statistical approach (Cox regression). For each patient, up to 20 general population controls without the studied mental disorders were matched by age (birth year) and sex. We adopted Cox regression to control for time to event, and to account for the potential impact of death as a competing event for arrest for IPV. In the current study, the rate of death during follow-up was higher among men with psychiatric diagnoses (1.6% to 7.1%) than among their matched general population controls (0.4% to 1.1%). Cox regression showed that, compared to general population controls, men with mental disorders were 3 to 11 times more likely to die during the follow-up. Thus, in our Cox regression, instead of omitting people who died during follow-up from the survival analyses, we treated "failure" from death as a censored observation, while "failure" from the outcome of interest (i.e., IPV) as an event. We report results from the Cox regression, with mental disorders as the predictor, and IPV against women after the diagnosis of a mental disorder as the outcome. We included family disposable income, single status, and immigrant status as confounders. Missing data are minimal in this study: less than 10% for income and 3% for single status across comparisons. No other data were missing.

To account for possible confounding by familial factors, we conducted additional analyses with unaffected, sex-matched full siblings of patients as controls. Unaffected full siblings were siblings without a diagnosis of the examined disorder but not necessarily without other mental disorders. For instance, when investigating the link between depression and IPV, the sibling comparisons were siblings without a diagnosis of depression but with or without substance use disorders or other psychiatric disorders. We compared unaffected full siblings of the patients with 20 age- and sex-matched general population controls with Cox regression. As in the models comparing patients and general population controls, we controlled for family disposable income, single status, and immigrant status and calculated hazard ratios of IPV against women for unaffected siblings of individuals with mental disorders.

Then, we compared the hazard ratios obtained in patient analyses to those obtained in sibling analyses using the ratio of hazard ratios (RHR). The RHR provides one way of accounting for familial factors including genetic and early family environmental factors. An RHR of 1 indicates that the risk of IPV against women in those with mental disorders is the same as the risk in their unaffected full siblings. That is, if there is an association between a mental disorder and IPV in the primary analysis, but the RHR is 1, then the association between the mental disorder and IPV is fully confounded by genetic and environmental factors shared by full siblings.

We conducted several additional sensitivity analyses. First, we compared the risk of arrest for IPV between psychiatric patients with and without comorbidity of alcohol use disorder, drug use disorder, or personality disorders, as these disorders are often comorbid with other psychiatric disorders and are associated with antisocial behaviors [34–37]. In addition, we conducted interaction analyses between mental disorders and comorbidity of these 3 disorders to further examine differences between groups in the Cox regression model. Second, to investigate confounding by substance use disorders, we adjusted associations between mental disorders and IPV for substance use disorders prior to the exposure. Third, we performed subgroup analysis by inpatient and outpatient diagnosis to examine group differences. Fourth, we examined the association between mental disorders and arrest for general domestic violence to

examine whether mental disorders were associated with IPV against women and with general domestic violence in a similar pattern in men. Fifth, to mitigate against reverse causality (because IPV might precipitate diagnoses of mental disorders), we ran separate analyses in a subgroup of individuals without a record of IPV before their diagnosis with a mental disorder. We used R statistical software in our analyses.

This study is reported as per the Strengthening the Reporting of Observational Studies in Epidemiology (STROBE) guidelines (S1 Checklist).

## Results

### Descriptive findings

We examined the risk of IPV against women by men in 9 diagnostic groups, with sample sizes ranging from 9,529 individuals with autism to 88,182 persons with depressive disorder. The average age at the beginning of follow-up (the year of receiving a diagnosis between 1998 and 2013) was 18 years for autism, 23 years for ADHD, and 30–34 years for the other mental disorders. Other characteristics are reported in Tables 1 and S1. The mean duration of follow-up across diagnoses ranged from 3.4 years to 4.8 years. The absolute rate of IPV perpetrated by men towards women ranged from 0.1% in individuals with autism to 2.1% in those with drug use disorder, from 0.2% to 0.8% among unaffected siblings, and from 0.1% to 0.4% in the matched general population controls (Table 2).

### Main results

When compared with general population controls (Table 2), all psychiatric diagnoses except autism were associated with an increased risk of IPV against women by men, with hazard ratios ranging from 1.5 (95% CI 1.3–1.7) to 7.7 (95% CI 7.2–8.3) ($p$-values $< 0.001$). Analyses comparing hazard ratios in patients (versus population controls) and unaffected siblings (versus population controls) showed that men with depressive disorder, anxiety disorder, alcohol use disorder, drug use disorder, ADHD, and personality disorders had higher risk of IPV against women than their unaffected full siblings, with RHR values ranging from 1.7 (95% CI 1.3–2.1) to 4.4 (95% CI 3.7–5.2) ($p$-values $< 0.001$) (Table 2; Fig 1). When comparing the hazard ratios from patient analyses to those from the sibling analyses, individuals with schizophrenia-spectrum disorders (RHR = 0.7 [95% CI 0.5–0.9], $p = 0.002$) and autism (RHR = 0.3 [95% CI 0.1–0.9], $p < 0.001$) had lower risks of IPV perpetration against women than their unaffected siblings. The covariates in the models were associated with IPV perpetration. Hazard ratios ranged from 1.7 to 2.4 for low (lowest tertile) family income, 1.2 to 2.1 for single status (except for in models testing the association of schizophrenia-spectrum disorders and bipolar disorder with IPV), and 3.5 to 6.4 for immigrant status across models (S2 Table).

### Sensitivity analyses

We conducted additional analyses by subgroups with and without comorbidity of alcohol use disorder, drug use disorder, and personality disorders (Tables 3 and S3). We found that the hazard ratio of IPV against women for men with mental disorders was increased with comorbid substance use disorders and personality disorders (except in autism). These group differences were supported by interaction effects between mental disorders (except for autism) and comorbidity (all $p$-values $< 0.001$, except for $p = 0.004$ for the interaction between schizophrenia-spectrum disorders and drug use disorder).

We conducted analyses with general domestic violence perpetrated by men as the outcome (S4 Table), and found that mental disorders were similarly associated with general domestic

**Table 1. Descriptive data for men with mental disorders, unaffected full siblings, and matched general population controls.**

| Mental disorder | Characteristic | Individuals with mental disorders[a] | General population controls | Unaffected full siblings |
|---|---|---|---|---|
| Schizophrenia-spectrum disorders | Sample size | 26,085 | 518,801 | 11,592 |
| | Follow-up start age | 32.7 (9.1) | 32.7 (9.1) | 31.0 (9.5) |
| | Low income | 17,152 (67.3%) | 150,569 (30.0%) | 3,975 (36.5%) |
| | Single status | 23,337 (90.8%) | 370,983 (73.3%) | 9,085 (79.6%) |
| | Born abroad | 6,623 (25.4%) | 63,579 (12.3%) | 1,263 (10.9%) |
| | Previous IPV | 372 (1.4%) | 1,776 (0.3%) | 62 (0.5%) |
| Bipolar disorder | Sample size | 12,065 | 239,388 | 5,758 |
| | Follow-up start age | 34.2 (10.2) | 34.1 (10.2) | 33.0 (10.2) |
| | Low income | 6,203 (53.2%) | 64,863 (28.6%) | 1,777 (32.7%) |
| | Single status | 9,696 (81.2%) | 166,445 (71.5%) | 4,230 (75.0%) |
| | Born abroad | 1,436 (11.9%) | 27,875 (11.6%) | 230 (4.0%) |
| | Previous IPV | 198 (1.6%) | 1,224 (0.5%) | 25 (0.4%) |
| Depressive disorder | Sample size | 88,182 | 1,719,318 | 36,453 |
| | Follow-up start age | 31.7 (10.8) | 31.7 (10.8) | 30.7 (10.9) |
| | Low income | 41,344 (49.2%) | 447,487 (27.8%) | 10,424 (31.2%) |
| | Single status | 72,276 (82.6%) | 1,266,415 (75.7%) | 28,006 (77.9%) |
| | Born abroad | 15,142 (17.2%) | 186,326 (10.8% | 2,066 (5.7%) |
| | Previous IPV | 1,334 (1.5%) | 6,631 (0.4%) | 165 (0.5%) |
| Anxiety disorder | Sample size | 60,355 | 1,195,303 | 28,962 |
| | Follow-up start age | 30.3 (11.1) | 30.3 (11.1) | 29.5 (10.7) |
| | Low income | 24,459 (44.2%) | 311,913 (29.0%) | 8,417 (32.2%) |
| | Single status | 49,790 (83.0%) | 909,608 (78.2%) | 22,906 (80.2%) |
| | Born abroad | 8,167 (13.5%) | 124,450 (10.4%) | 1,985 (6.9%) |
| | Previous IPV | 790 (1.3%) | 4,583 (0.4%) | 134 (0.5%) |
| Alcohol use disorder | Sample size | 82,731 | 1,643,539 | 37,885 |
| | Follow-up start age | 31.1 (11.2) | 31.0 (11.2) | 29.4 (11.2) |
| | Low income | 40,172 (51.3%) | 405,390 (26.4%) | 10,431 (30.9%) |
| | Single status | 73,707 (89.9%) | 1,226,776 (76.5%) | 30,531 (81.5%) |
| | Born abroad | 10,080 (12.2%) | 184,520 (11.2%) | 2,251 (5.9%) |
| | Previous IPV | 1,744 (2.1%) | 4,743 (0.3%) | 149 (0.4%) |
| Drug use disorder | Sample size | 57,901 | 1,151,306 | 24,116 |
| | Follow-up start age | 30.6 (10.3) | 30.5 (10.3) | 29.6 (10.5) |
| | Low income | 33,745 (59.9%) | 334,162 (30.1%) | 7,845 (35.3%) |
| | Single status | 51,976 (91.0%) | 883,380 (78.6%) | 19,463 (81.8%) |
| | Born abroad | 9,783 (16.9%) | 126,675 (11.0%) | 2,302 (9.5%) |
| | Previous IPV | 1,360 (2.3%) | 3,440 (0.3%) | 117 (0.5%) |
| ADHD | Sample size | 49,327 | 976,123 | 22,576 |
| | Follow-up start age | 23.3 (11.5) | 23.3 (11.4) | 24.1 (11.2) |
| | Low income | 20,212 (45.5%) | 234,967 (26.9%) | 6,603 (31.6%) |
| | Single status | 46,321 (94.2%) | 839,175 (88.1%) | 19,795 (88.7%) |
| | Born abroad | 3,923 (8.0%) | 70,768 (7.3%) | 965 (4.3%) |
| | Previous IPV | 804 (1.6%) | 2,635 (0.3%) | 96 (0.4%) |
| Autism | Sample size | 9,529 | 186,017 | 4,818 |
| | Follow-up start age | 17.7 (10.4) | 18.0 (10.3) | 19.6 (10.3) |
| | Low income | 3,116 (39.5%) | 41,453 (26.6%) | 1,255 (30.1%) |
| | Single status | 9,432 (99.3%) | 171,504 (94.0%) | 4,501 (94.2%) |
| | Born abroad | 796 (8.4%) | 10,838 (5.8%) | 261 (5.4%) |

*(Continued)*

**Table 1.** (Continued)

| Mental disorder | Characteristic | Individuals with mental disorders[a] | General population controls | Unaffected full siblings |
|---|---|---|---|---|
| | Previous IPV | 18 (0.2%) | 271 (0.1%) | 9 (0.2%) |
| Personality disorders | Sample size | 19,850 | 394,367 | 9,128 |
| | Follow-up start age | 33.4 (8.8) | 33.4 (8.8) | 31.9 (9.5) |
| | Low income | 13,941 (71.2%) | 117,542 (30.7%) | 3,313 (38.0%) |
| | Single status | 17,740 (90.4%) | 279,429 (72.8%) | 7,100 (78.9%) |
| | Born abroad | 3,336 (16.8%) | 48,081 (12.2%) | 688 (7.5%) |
| | Previous IPV | 647 (3.3%) | 1,606 (0.4%) | 62 (0.7%) |

Values are given as *n* (%), except for follow-up start age, which is given as mean (standard deviation).

[a]Including all patients with or without unaffected full siblings.

ADHD, attention deficit hyperactivity disorder; IPV, intimate partner violence.

violence. Hazard ratios ranged from 1.6 (95% CI 1.4–1.9) to 7.0 (95% CI 6.6–7.5) (*p*-values < 0.001) when comparing individuals with mental disorders to population controls; substance use disorders showed the highest hazard ratios (HRs > 6.2, *p*-values < 0.001). Comparisons with hazard ratios in unaffected full siblings (versus general population controls) also showed similar patterns, with RHRs ranging from 1.4 (95% CI 1.1–1.8) to 3.7 (95% CI 3.0–4.4) (*p*-values < 0.001).

**Table 2. Risks of IPV against women in men with mental disorders and their unaffected siblings compared to general population controls, and also in men with mental disorders compared to their unaffected siblings (as ratio of adjusted hazard ratios [RHR]).**

| Mental disorder | Individuals with mental disorders who perpetrated IPV | | | | | | | | | Unaffected full siblings who perpetrated IPV | | | | | | | RHR | CI | *p*-Value |
|---|---|---|---|---|---|---|---|---|---|---|---|---|---|---|---|---|---|---|---|
| | *n* (%) | % perpetrated IPV in controls | Follow-up years (SD) | PAR (%) | cHR | CI | aHR | CI | *p*-Value | *n* (%) | % perpetrated IPV in controls | cHR | CI | aHR | CI | *p*-Value | | | |
| Schizophrenia-spectrum disorders | 209 (0.8%) | 0.4% | 4.0 (3.1) | 4.6% | 1.9 | 1.7–2.2 | 1.5 | 1.3–1.7 | <0.001 | 93 (0.8%) | 0.3% | 2.7 | 2.2–3.4 | 2.2 | 1.8–2.7 | <0.001 | 0.7 | 0.5–0.9 | 0.002 |
| Bipolar disorder | 60 (0.5%) | 0.3% | 3.4 (3.0) | 3.1% | 2.2 | 1.7–2.8 | 2.2 | 1.7–2.8 | <0.001 | 13 (0.2%) | 0.2% | 1.3 | 0.7–2.3 | 1.2 | 0.6–2.2 | 0.600 | 1.8 | 0.9–3.7 | 0.088 |
| Depressive disorder | 705 (0.8%) | 0.2% | 3.7 (2.9) | 12.8% | 3.3 | 3.1–3.6 | 2.9 | 2.7–3.2 | <0.001 | 73 (0.2%) | 0.2% | 1.3 | 1.1–1.7 | 1.2 | 0.9–1.5 | 0.150 | 2.4 | 1.9–3.2 | <0.001 |
| Anxiety disorder | 362 (0.6%) | 0.2% | 3.7 (3.0) | 8.8% | 2.5 | 2.3–2.8 | 2.5 | 2.2–2.7 | <0.001 | 87 (0.3%) | 0.2% | 1.7 | 1.4–2.2 | 1.5 | 1.2–1.9 | <0.001 | 1.7 | 1.3–2.1 | <0.001 |
| Alcohol use disorder | 1,406 (1.7%) | 0.3% | 4.6 (3.4) | 18.3% | 6.1 | 5.8–6.5 | 7.0 | 6.6–7.5 | <0.001 | 152 (0.4%) | 0.2% | 1.7 | 1.4–2.0 | 1.6 | 1.4–1.9 | <0.001 | 4.4 | 3.7–5.2 | <0.001 |
| Drug use disorder | 1,216 (2.1%) | 0.3% | 4.7 (3.4) | 22.3% | 7.2 | 6.7–7.7 | 7.7 | 7.2–8.3 | <0.001 | 121 (0.5%) | 0.2% | 2.4 | 2.0–2.9 | 2.1 | 1.7–2.5 | <0.001 | 3.7 | 3.0–4.5 | 0.028 |
| ADHD | 296 (0.6%) | 0.1% | 3.8 (3.2) | 19.4% | 5.1 | 4.5–5.7 | 6.4 | 5.5–7.4 | <0.001 | 45 (0.2%) | 0.1% | 2.1 | 1.5–2.8 | 2.1 | 1.5–2.8 | <0.001 | 3.1 | 2.2–4.3 | <0.001 |
| Autism | 10 (0.1%) | 0.1% | 4.8 (3.3) | 0.2% | 0.5 | 0.2–1.2 | 0.7 | 0.3–1.6 | 0.380 | 10 (0.2%) | 0.1% | 2.1 | 1.2–3.9 | 2.2 | 1.2–3.9 | 0.010 | 0.3 | 0.1–0.9 | <0.001 |
| Personality disorders | 337 (1.7%) | 0.4% | 4.4 (3.3) | 13.5% | 4.8 | 4.3–5.4 | 4.3 | 3.8–4.9 | <0.001 | 64 (0.7%) | 0.2% | 3.0 | 2.3–3.9 | 2.5 | 1.9–3.3 | <0.001 | 1.7 | 1.3–2.3 | <0.001 |

Each individual with a mental disorder and his unaffected full sibling were compared with 20 age- and sex-matched general population controls. Adjusted hazard ratio analyses were adjusted for family income, single status, and immigrant status.

ADHD, attention deficit hyperactivity disorder; aHR, adjusted hazard ratio; cHR, crude hazard ratio (not adjusted for any covariates); CI, confidence interval; IPV, intimate partner violence; PAR (%), population attributable risk percent; SD, standard deviation.

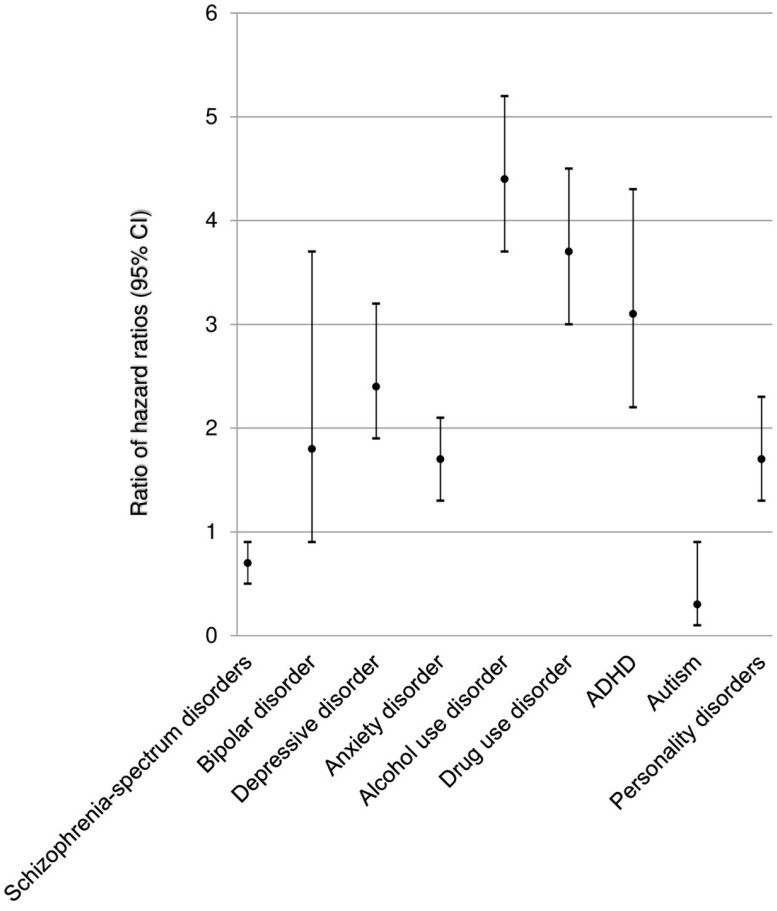

**Fig 1. Ratio of hazard ratios (men with mental disorders versus siblings) of intimate partner violence against women.** The ratio of hazard ratios is the hazard ratio for individuals with mental disorders versus 20 age- and sex-matched general population controls divided by the hazard ratio for unaffected siblings versus 20 age- and sex-matched general population controls. The bars represent the 95% confidence intervals. ADHD, attention deficit hyperactivity disorder.

In addition, individuals with an inpatient psychiatric diagnosis in general showed a higher hazard ratio than those with an outpatient diagnosis, particularly for depressive, anxiety, and drug use disorders (S5 Table).

Furthermore, we conducted a series of other sensitivity analyses. In one set of analyses, we removed individuals with a history of IPV, and in another we adjusted for alcohol and drug use disorders prior to a psychiatric diagnosis. We also tested models without adjusting for any covariates. All of these analyses did not materially change the hazard ratios of IPV perpetration among men with mental disorders (Tables 2, S6 and S7).

## Discussion

We examined the risk of IPV against women perpetrated by men with 9 psychiatric diagnoses in a Swedish population-based study over 1998–2013. The sample sizes of diagnostic groups ranged from 9,529 individuals (with autism) to 88,182 (with depressive disorder). When compared to the general population, we found that men with mental disorders, apart from those with autism, were more likely to perpetrate IPV against women. These associations remained after adjustment for familial confounding, apart from the low-prevalence disorders (likely due

**Table 3. Hazard ratio of intimate partner violence against women in men with mental disorders by psychiatric comorbidity.**

| Mental disorder | Comorbidity of alcohol use disorder | | | | | | | | Comorbidity of drug use disorder | | | | | | | | Comorbidity of personality disorder | | | | | | | |
|---|---|---|---|---|---|---|---|---|---|---|---|---|---|---|---|---|---|---|---|---|---|---|---|---|
| | Yes | | | | No | | | | Yes | | | | No | | | | Yes | | | | No | | | |
| | n | aHR | CI | p | n | aHR | CI | p | n | aHR | CI | p | n | aHR | CI | p | n | aHR | CI | p | n | aHR | CI | p |
| Schizophrenia-spectrum disorders | 5,525 | 3.1 | 2.4–3.9 | <0.001 | 20,560 | 1.1 | 0.9–1.3 | 0.340 | 7,171 | 3.3 | 2.6–4.1 | <0.001 | 18,914 | 0.9 | 0.8–1.1 | 0.540 | 4,424 | 3.4 | 2.6–4.4 | <0.001 | 21,661 | 1.1 | 0.9–1.3 | 0.330 |
| Bipolar disorder | 2,960 | 5.2 | 3.5–7.6 | <0.001 | 9,105 | 1.3 | 0.9–1.8 | 0.200 | 2,543 | 5.3 | 3.4–8.2 | <0.001 | 9,522 | 1.5 | 1.1–2.1 | 0.021 | 1,825 | 4.8 | 2.7–8.6 | <0.001 | 10,240 | 1.9 | 1.4–2.5 | <0.001 |
| Depressive disorder | 16,174 | 7.3 | 6.4–8.5 | <0.001 | 72,008 | 1.9 | 1.7–2.1 | <0.001 | 13,397 | 6.6 | 5.6–7.7 | <0.001 | 74,785 | 2.2 | 2.0–2.4 | <0.001 | 7,437 | 4.9 | 4.0–6.0 | <0.001 | 80,745 | 2.7 | 2.4–2.9 | <0.001 |
| Anxiety disorder | 7,962 | 7.6 | 6.1–9.4 | <0.001 | 52,393 | 1.7 | 1.5–2.0 | <0.001 | 7,463 | 8.5 | 6.9–10.4 | <0.001 | 52,892 | 1.6 | 1.4–1.9 | <0.001 | 2,920 | 4.4 | 3.2–6.1 | <0.001 | 57,435 | 2.3 | 2.0–2.6 | <0.001 |
| Alcohol use disorder | | | | | | | | | 21,275 | 10.4 | 9.3–11.5 | <0.001 | 61,456 | 5.6 | 5.1–6.1 | <0.001 | 6,125 | 11.2 | 9.3–13.5 | <0.001 | 76,606 | 6.6 | 6.2–7.1 | <0.001 |
| Drug use disorder | 21,334 | 11.1 | 9.9–12.3 | <0.001 | 36,567 | 5.6 | 5.1–6.2 | <0.001 | | | | | | | | | 7,220 | 11.1 | 9.5–13.0 | <0.001 | 50,681 | 7.0 | 6.5–7.6 | <0.001 |
| ADHD | 8,527 | 12.6 | 9.8–16.3 | <0.001 | 40,800 | 4.5 | 3.8–5.5 | <0.001 | 10,791 | 11.3 | 9.1–14.1 | <0.001 | 38,536 | 4.2 | 3.4–5.1 | <0.001 | 4,391 | 12.5 | 9.2–17.0 | <0.001 | 44,936 | 5.2 | 4.4–6.2 | <0.001 |
| Autism | 419 | 8.1 | 2.0–32.6 | 0.003 | 9,110 | 0.4 | 0.1–1.2 | 0.085 | 421 | 7.3 | 1.5–36.0 | 0.014 | 9,108 | 0.3 | 0.1–1.1 | 0.081 | 364 | 3.3 | 0.5–20.7 | 0.200 | 9,165 | 0.5 | 0.2–1.4 | 0.170 |
| Personality disorders | 6,144 | 6.7 | 5.5–8.3 | <0.001 | 13,706 | 3.1 | 2.6–3.7 | <0.001 | 7,221 | 6.4 | 5.3–7.7 | <0.001 | 12,629 | 2.8 | 2.3–3.5 | <0.001 | | | | | | | | |

Each individual with a mental disorder was compared with 20 age- and sex-matched general population controls. aHR analyses were adjusted for family income, single status, and immigrant status.

ADHD, attention deficit hyperactivity disorder; aHR, adjusted hazard ratio; CI, confidence interval.

to lack of statistical power to show differences). Men with alcohol and drug use disorders had the highest risks (7- to 8-fold increased risks) compared with general population controls, and those with ADHD and personality disorders were also consistently at an increased risk across models. Furthermore, the comorbidity of substance use disorders and personality disorders increased the risk of IPV against women in men with all investigated psychiatric diagnoses.

Our findings underscore that substance use disorders are the primary diagnoses with the highest relative risk among all studied disorders for the risk of IPV perpetration, and that substance use disorder comorbidity increases the risk of IPV perpetration for other mental disorders. Alcohol and drug use disorders decrease an individual's inhibition, which in turn can lead to the use of violence to solve conflicts in intimate relationships [38]. People with mental disorders are also likely to use alcohol and drugs as coping strategies to deal with difficult symptoms associated with their illnesses [39,40]. Therefore, alcohol and drug use disorders could be underlying mechanisms linking other mental disorders to later IPV perpetration, in addition to being strong independent predictors themselves. Overall, our findings suggest that prevention and intervention programs should prioritize assessment of risk of IPV in men with diagnosis of substance use disorders, especially because these disorders are treatable [41].

Furthermore, the comorbidity of substance use disorders was associated with a substantially increased risk of IPV perpetration in all the other mental disorders, including autism, which did not show a higher risk on its own compared to general population controls. These results could help reduce the stigma around IPV perpetration in mental disorders in general, as IPV risk was much lower without comorbidity of substance use disorders. In addition, the findings provide an important preventative target for clinicians working in adult mental health services, who may not be including risk to intimate partners as part of their risk assessments nor focusing on the risk patients may pose in the context of drug and alcohol misuse (which is more common in individuals with mental disorders than in the general population) [42].

We found that schizophrenia-spectrum disorders showed higher risk of IPV perpetration than general population controls. However, individuals with these disorders did not show higher risk than their unaffected full siblings, although this may reflect low statistical power. This result, although needing further replication, contrasts with those from studies reporting links between schizophrenia-spectrum disorders and general violence [43]. In addition, research that showed an association between psychosis and domestic homicides also reported that perpetrators with symptoms at the time of offense were less likely than perpetrators without symptoms to have previous violence convictions [44]. It is possible that common symptoms of schizophrenia-spectrum disorders such as paranoid ideation are associated with general violence but not necessarily with violence against intimate partners [45].

It is important to note that the results for autism and schizophrenia-spectrum disorders were different. That is, autism was associated with a lower risk of IPV both in the general population comparisons and sibling comparisons. However, schizophrenia-spectrum disorders were associated with a higher risk of IPV in the general population comparisons but not in the sibling analyses. Familial factors might explain this association. It is possible that unaffected siblings may not have a clinical diagnosis of a schizophrenia-spectrum disorder but may still have underlying cognitive impairments, which may lead to IPV in both affected and unaffected full siblings. Moreover, it could be that individuals with autism are less likely to have intimate partners and thus have less opportunity for violence against partners. In addition, those who have partners might present with less severe symptoms of autism.

We found a higher risk of IPV perpetration among individuals with an inpatient diagnosis than with an outpatient diagnosis for 3 psychiatric diagnoses (depressive, anxiety, and drug use disorders). This suggests that the links between these mental disorders and IPV might

function in a dose–response pattern, as inpatients can be assumed to have more severe underlying disorders.

Low income was associated with IPV perpetrated by men towards women. This finding is consistent with existing research on the link between financial distress and increased IPV [46]. In addition, men who were not married were more likely to commit IPV against women. This could mean that, on average, marriage implies a more stable and committed relationship than unmarried partnership, and thus is associated with reduced IPV risks. Furthermore, we found that immigrant status (being born outside of Sweden) was associated with a higher risk of IPV against women, which may be explained by cultural differences [2].

Overall, we have shown that mental disorders, particularly substance use disorders, personality disorders, and ADHD, are associated with an increased risk of IPV perpetration. Therefore, treatment of these disorders could potentially reduce the risk in these groups, especially as evidence-based interventions exist [47–50]. For example, it has been reported that among ADHD patients receiving medication, a significant reduction of criminality rate is observed [51]. Furthermore, integrated interventions for mental disorders and IPV may be particularly helpful. This approach is supported by a randomized controlled trial of cognitive behavioral therapy that reduced both the symptoms of substance use disorders and IPV among male offenders [52].

Although our study is observational and causality cannot be inferred, if causality was assumed, then population attributable risk percentages could be interpreted as the maximum possible impact that fully treating a disorder would have on IPV—these ranged from 0.2% for autism to 22.3% for drug use disorders. Treating substance misuse, common deficits such as affect regulation, and specific symptoms of mental disorders might be an important step to prevent IPV against women in some men with psychiatric diagnoses. These findings also suggest that to reduce men's IPV against women, other modifiable risk factors in addition to mental disorders need to be considered. More specifically, other individual risk factors of men's IPV against women include comorbidity of substance use disorders, as showed in this study, and stressful life events, such as previous victimization and witnessing domestic violence during childhood [53,54]. Apart from developmental history and current characteristics of individuals, the WHO ecological framework highlights that environmental factors including gender disadvantage (e.g., in education and employment), structural factors, and characteristics of the relationship could also contribute to IPV against women [1,55,56]. It is currently unclear how factors at the individual level interact with associations at the relationship, community, and societal levels. Future research is necessary to clarify this.

Our findings also highlight the need for examining underlying mechanisms. In addition to providing treatment for common deficits and specific core symptoms of mental disorders, it is important to examine factors at the relationship level. It is likely that individuals with mental disorders selectively end up in abusive intimate partnerships, which could lead to reactive violence towards partners [57]. Moreover, there has been evidence of assortative mating (or nonrandom mating) within and across major mental disorders such as substance use disorders, schizophrenia, depression, and ADHD [58], which might increase the risk of IPV perpetration due to cognitive and social impairments in both partners. Empirical studies are needed to examine potential mediators linking mental disorders to IPV perpetration.

Our study has several strengths. First, we used a longitudinal research design, which accounts for the temporal sequence between mental disorders and IPV perpetration. Second, we tested the associations between mental disorders and IPV perpetration in a population-based sample, which increases the generalizability of the findings. In addition, using arrest for IPV as the outcome means that the findings may be more reliable, as self-report may be more prone to cultural biases [59]. Third, although we cannot demonstrate a causal relationship, the

criminal coding in the National Crime Register enabled us to retrieve data on this specific type of violent perpetration in men, which is not recorded separately in many other countries. Furthermore, using arrest data from this national register enabled analyses with sufficient statistical power to study these associations more precisely than previous work, and the arrest data could be linked accurately with healthcare and family registers—a comparable interview-based study would be very large and expensive, and may not be possible (especially in finding and interviewing siblings). The Multi-Generation Register allowed us to conduct full sibling comparisons to control for shared familial factors that might contribute to both mental disorders and IPV perpetration [27]. Fourth, we also conducted a series of sensitivity analyses, such as analyses removing individuals with a history of IPV, subgroup analyses of persons with comorbidity of substance use disorders and personality disorders, and analyses adjusting for alcohol and drug use disorders prior to a psychiatric diagnosis. We found no material differences in hazard ratios between results from these sensitivity analyses and those from the main analyses. These complementary methods enabled us to provide more precision in estimating the link between mental disorders and IPV. These strengths help overcome limitations of prior studies based on selected samples [60].

Several limitations should be noted. First, we used arrest for IPV as the outcome. The absolute rate of IPV perpetration arrests ranged from 0.1% in men with autism to 2.1% in men with alcohol use disorder over an average follow-up of 3.4 to 4.8 years. As it has been widely recognized that victims of partner abuse tend to not report the abuse to the police, and many men might have already perpetrated IPV prior to the age of 15 years (the age of inclusion in the crime register), not all IPV is captured by this approach [61], and our findings are specific to more severe forms of IPV perpetration that lead to arrest and likely have significant negative consequences for victims, such as serious morbidity and in rare cases mortality. However, other research has shown that the degree of underreporting of violence is similar for violence perpetrated by patient groups and by the general population [62]. Therefore, even though we only captured a subset of IPV perpetrations, the relative risk estimates in the patient analyses (patients versus general population controls) and sibling analyses (siblings versus general population controls) should not be significantly affected. Second, there could be a selection bias against individuals who are in a vulnerable position, such as those with low socioeconomic status being more likely to be arrested for IPV [63]. This may have inflated the prevalence of IPV among certain diagnostic groups. The risk estimates therefore might be overestimated, although our adjustment for income, use of sibling comparisons, and sensitivity analyses excluding individuals arrested for IPV before diagnosis most likely mitigated this potential bias. Third, our exposure was psychiatric diagnosis. As many individuals with a mental disorder do not get a formal diagnosis, our patients represent the more severe cases (but those who are accessing services). For example, in this study, individuals with substance use disorders were those with a diagnosis recorded in official patient registries. Most people with a substance use disorder might never get a diagnosis. Thus, our sample represented a group of people with more severe substance use problems but with the advantage that they are accessing services and therefore can be further assessed and treated. Fourth, as some of the disorders investigated were relatively rare (schizophrenia-spectrum disorders, bipolar disorders), the sibling comparisons were likely underpowered to demonstrate differences. Fifth, previous studies have suggested that borderline and antisocial personality disorders are more likely to be associated with IPV than dependent personality disorders [64]. However, due to lack of data on the diagnostic validity of specific personality disorders, we did not investigate links between individual personality disorders and IPV. Finally, the study was done in one country. However, as the prevalences of IPV against women (23.2% in Sweden versus 25.4% in Europe overall) and mental

disorders in Sweden are similar to those in other high-income countries [65–69], our findings are likely generalizable to other high-income countries.

## Conclusions

In summary, we examined the link between mental disorders and later IPV using a large population-based cohort of male IPV perpetrators and, to our knowledge for the first time, compared the risks in men with mental disorders to those in unaffected siblings, to account for genetic and family environmental factors. Our results suggest longitudinal associations between many mental disorders, particularly substance use disorders, ADHD, and personality disorders, and IPV against women by men. Substance use disorders as a primary diagnosis were associated with the highest risk of IPV perpetration among the studied psychiatric diagnoses, and comorbid substance use disorders were associated with an increased likelihood of IPV perpetration in all of the other disorders examined. Our findings suggest that prioritizing the development of services to assess IPV perpetration among men with substance use disorders may help to reduce the risk of IPV against women.

## Supporting information

**S1 Checklist. STROBE statement—Checklist of items that should be included in reports of cohort studies.**
(DOC)

**S1 Table. Descriptive data for risk factors in unaffected full siblings and matched general population controls.**
(DOCX)

**S2 Table. Hazard ratio (HR) for confounders in the models comparing risk of IPV against women in men with mental disorders with that in their matched general population controls.**
(DOCX)

**S3 Table. Crude hazard ratio (cHR) of IPV against women in men with mental disorders by psychiatric comorbidity.**
(DOCX)

**S4 Table. Hazard ratio (HR) and ratio of hazard ratios (RHR) of general domestic violence in men with mental disorders and their unaffected full siblings.**
(DOCX)

**S5 Table. Hazard ratio (HR) of IPV against women in men with inpatient versus outpatient diagnosis of mental disorder.**
(DOCX)

**S6 Table. Hazard ratio (HR) and ratio of hazard ratios (RHR) of IPV against women in men with mental disorders and their unaffected full siblings after excluding individuals with a previous IPV arrest.**
(DOCX)

**S7 Table. Adjusted hazard ratio (aHR) and ratio of hazard ratios (RHR) of IPV against women in men with mental disorders after adjustment of prior alcohol and drug use disorders.**
(DOCX)

## Author Contributions

**Conceptualization:** Rongqin Yu.

**Formal analysis:** Rongqin Yu, Alejo J. Nevado-Holgado.

**Funding acquisition:** Seena Fazel.

**Investigation:** Rongqin Yu, Yasmina Molero.

**Methodology:** Rongqin Yu, Alejo J. Nevado-Holgado, Brian M. D'Onofrio, Henrik Larsson, Louise M. Howard, Seena Fazel.

**Project administration:** Rongqin Yu.

**Supervision:** Seena Fazel.

**Writing – original draft:** Rongqin Yu.

**Writing – review & editing:** Rongqin Yu, Yasmina Molero, Brian M. D'Onofrio, Henrik Larsson, Louise M. Howard, Seena Fazel.

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
