## [Decision Letter · Decision Letter 0]

9 Sep 2019

Dear Dr. Fazel,

Thank you very much for submitting your manuscript "Mental disorders and men’s intimate partner violence against women: a population-based longitudinal study" (PMEDICINE-D-19-02530) for consideration at PLOS Medicine. 

Your paper was evaluated by a senior editor and discussed among all the editors here. It was also discussed with an academic editor with relevant expertise, and sent to three independent reviewers, including a statistical reviewer. The reviews are appended at the bottom of this email and the accompanying attachment from Reviewer 2 can be seen via the link below:

[LINK]

In light of these reviews, I am afraid that we will not be able to accept the manuscript for publication in the journal in its current form, but we would like to consider a revised version that addresses the reviewers' and editors' comments. Obviously we cannot make any decision about publication until we have seen the revised manuscript and your response, and we plan to seek re-review by one or more of the reviewers. 

We expect to receive your revised manuscript by Sep 23 2019 11:59PM. Please email us (plosmedicine@plos.org) if you have any questions or concerns.

We look forward to receiving your revised manuscript. 

Sincerely,

Caitlin Moyer, Ph.D.

Associate Editor 

PLOS Medicine

plosmedicine.org

1.Did your study have a prospective protocol or analysis plan? Please state this (either way) early in the Methods section.

c) In either case, changes in the analysis—including those made in response to peer review comments—should be identified as such in the Methods section of the paper, with rationale.

2. Thank you for your note that study data are available from “...Karolinska Institute Data Access for researchers who meet the criteria for access to confidential data.”

However, PLOS Medicine requires that the de-identified data underlying the specific results in a published article be made available, without restrictions on access, in a public repository or as Supporting Information at the time of article publication, provided it is legal and ethical to do so. Please see the policy at: 

http://journals.plos.org/plosmedicine/s/data-availability

and FAQs at: 

http://journals.plos.org/plosmedicine/s/data-availability#loc-faqs-for-data-policy

3. Abstract: Methods and findings: * Please ensure that all numbers presented in the abstract are present and identical to numbers presented in the main manuscript text. There are typos/missing commas in the numbers of the study participants- please fix/add commas.

4. Abstract: Methods and findings: Please quantify the main results (with 95% CIs and p values).

5. Abstract: Methods and findings: In the last sentence of the Abstract Methods and Findings section, please describe the main limitation(s) of the study's methodology.

6. Author Summary: At this stage, we ask that you include a short, non-technical Author Summary of your research to make findings accessible to a wide audience that includes both scientists and non-scientists. The Author Summary should immediately follow the Abstract in your revised manuscript. This text is subject to editorial change and should be distinct from the scientific abstract. Please see our author guidelines for more information: https://journals.plos.org/plosmedicine/s/revising-your-manuscript#loc-author-summary

7. Introduction: Please conclude the Introduction with a clear description of the study question or hypothesis. A clear description of the study’s main objective(s) is missing.

8. Introduction: Please consider revising the final sentence (comment on largest size and statement of primacy), at least consider qualifying it by including the phrase “to date” in your assertion that this is “the largest epidemiological study” in case this status changes in the future.

9. Introduction (and Abstract): Please define the abbreviation “ADHD” at the instance of first use.

10. Methods and Results: Please provide the actual numbers of events for the outcomes, not just the absolute rates. Specifically, provide the actual numbers associated with the rates of IPV for each population (Table 2 data). It is not clear where the absolute rates of IPV are provided for the population controls. Please specify in the first paragraph of the results section where these results are presented.

11. Methods and Results: Please provide p values for comparisons of hazard ratios in the text, as well as in tables 2 and 3, and appendixes 1, 3, 4, and 5. Please specify the statistical test used for comparisons.

12. Methods and Results: Please provide the p values for comparisons between groups. Specifically, in the description of the sensitivity analyses (“These group differences were supported by interaction effects between mental disorders (except for autism) and comorbidity (p’s ≤ .01).”) please specify the p value (unless p<0.001) and the statistical test used.

13. Methods and Results: Please provide the name(s) of the institutional review board(s) that provided ethical approval.

14. Methods and Results: Please specify whether informed consent was written or oral, or the conditions permitting the waiver of informed consent.

15. Methods and Results, and Discussion: In the first paragraph of the results, the number of individuals with depressive disorders is missing a comma. Similarly, a comma is missing from the number of individuals reported in the first paragraph of the discussion. Please edit throughout.

16. Discussion: Please revise the following sentence: “Furthermore, as the comorbidity of substance use disorders substantially increased the risk of IPV perpetration in all the other mental disorders, including autism which did not show a higher risk when compared to general population controls, these findings could help reducing the stigma around IPV perpetration in mental disorders in general as their higher risk is largely due to substance use disorders.” 

Specifically, your study is observational and therefore causality cannot be inferred. Please remove language that implies causality, such as “...as their higher risk is largely due to substance use disorders.” This statement implies causality. Refer to associations instead.

17. Discussion/Conclusion: Please avoid assertions of primacy ("We report for the first time....") and greatest size. Specifically, please revise the following sentence: “In summary, we examined the link between mental disorders and later IPV using the largest sample of IPV perpetrators and for the first time compared to risks in unaffected siblings to account for genetic and family environmental factors.”

18. Discussion/Conclusion: The statement “...and comorbid substance use disorders increased the risk of IPV against women in all of the other disorders examined…” implies causality. Your study is observational and therefore causality cannot be inferred. Please revise and refer to associations instead.

19. Table 2, Table 3, Figure 1, Appendices 1,3, 4, and 5: Please define the abbreviation “CI” in the legend.

20. All Tables and Figures: Please define the abbreviation “ADHD” in the legend.

21. Table 1, and Appendix 2: Please clarify which variables are N (%) and which variables are mean (SD).

22. Please ensure that the study is reported according to the STROBE guideline, and include the completed STROBE checklist as Supporting Information. When completing the checklist, please use section and paragraph numbers, rather than page numbers. Please add the following statement, or similar, to the Methods: "This study is reported as per the Strengthening the Reporting of Observational Studies in Epidemiology (STROBE) guideline (S1 Checklist)."

Comments from the reviewers:

Reviewer #1: This is a well-conducted study on the associations between mental disorders and men's intimate partner violence against women using population data. The study design, datasets, statistical methods and analyses, presentation (tables and figures) and interpretation of results are mostly adequate and of a good standard. However, still a few statistical issues needing attention.

1) In the statistical analyses of the Methods section, it says 'We compared patients' unaffected full siblings with 20 age and gender- matched general population controls with matched conditional logistic

regression'. What does this mean and what is it for? We have some odd ratios here but never appeared anywhere in the paper as only Hazard Ratios were applied throughout the paper.

2) Competing risk. Cox models were applied in the paper to assess the risks, and the outcome is the men's IPV against women other than all-cause mortality, therefore death could be a competing risk in the survival analysis. Can authors elaborate what the death rates are in these cohort? What's its impact on survival analysis in terms of competing risk?

3) As all the cox models were adjusted for potential confounders, we would like to see the influence/impact of these confounders in the analyses, such as the impact of family income, single status, and immigrant status. We didn't see these confounders presented and discussed in the results or discussion sections.

Reviewer #2: This is a very good and relevant paper

Reviewer #3: The authors examine the association between ICD-9 mental disorders and official arrests for IPV in a very large Swedish population cohort. Results indicate that the presence of almost any disorder increases risk for IPV arrests, with substance use disorders and ADHD having the strongest associations with IPV outcomes.

There are many positives to this important paper. The manuscript is efficiently written and very well-organized. The statistical approach is excellent, and the manner with which they address important confounding variables is a notable strength. The results are clearly presented, and conclusions are closely aligned with their findings. The paper truly stands apart from comparable studies that have been published regarding the mental illness-IPV link in terms of sample, data analysis/statistical controls, presentation of findings, and overall quality.

If there is one area that is below the generally high quality seen in the overall paper and is in need of further narrative attention, it is the authors' discussion of the etiological links between mental illness and IPV (including the role of substance use in IPV perpetration). There was no discussion of how or why mental illness might be linked to IPV in the Introduction, and the authors' coverage of this topic in the Discussion was severely lacking in breadth and depth. This is an exceedingly important issue in the IPV field in particular, as large sections of the field steadfastly refuse to acknowledge the role of mental illness in any form (e.g., personality traits, specific diagnoses, even alcohol use) as being causally related to IPV perpetration. This resistance, often rooted in protofeminist models of patriarchal socialization, consider such factors as excuses rather than the actual causes of IPV, which are presumed to be rooted in acceptance of personal responsibility. The main point is that there should be no a priori presumption that a large section of readers of this article will accept the very idea of this research, much less the actual findings. Therefore, more careful attention needs to be paid to specific etiological models of how, why, and which mental illnesses are connected to IPV perpetration. While there are a few sentences devoted to some potential mechanisms on p. 14, the authors state that (a) emotion regulation might be involved in emotion disorders (of course); (b) that the psychoactive properties of alcohol might relate to aggression (of course); and (c) that two subtypes of IPV might be important considerations (even though the subtype construct has largely fallen out of scientific favor). The authors are encouraged to be more mindful in their discussion of relevant theory and to more conscientiously discuss the actual and potential mechanisms associated with these models that link mental disorder to IPV perpetration.

[LINK]

---

## [Decision Letter · Decision Letter 1]

28 Oct 2019

Dear Dr. Fazel,

Thank you very much for re-submitting your manuscript "Mental disorders and intimate partner violence perpetrated by men towards women: a population-based longitudinal study" (PMEDICINE-D-19-02530R1) for review by PLOS Medicine.

I have discussed the paper with my colleagues and the academic editor and it was also seen again by 2 reviewers. I am pleased to say that provided the remaining editorial and production issues are dealt with we are planning to accept the paper for publication in the journal.

[LINK]

We look forward to receiving the revised manuscript by Nov 04 2019 11:59PM. 

Sincerely,

Caitlin Moyer, Ph.D.

Associate Editor 

PLOS Medicine

plosmedicine.org

Requests from Editors:

1.Title: Please revise the title to: “Mental disorders and intimate partner violence perpetrated by men towards women: a Swedish population-based longitudinal study”

2.Data Availability: Thank you for providing the weblink to request access to the individual level data. However, this link goes to a very general page. If you can provide a more specific location or contact information where readers can request access to the data, that would be helpful. 

3.Abstract: Background: In the final sentence, please avoid any misleading implications that the sibling comparison controlled for all genetic and family environmental factors. We suggest removing the word “all” from the sentence.

4.Abstract: Methods and Findings: Please include the years during which the study took place, and the length of follow up.

5. Abstract: Methods and Findings: Please revise the final sentence to: “A limitation of our study is that our analysis was restricted to instances of IPV leading to arrest, suggesting that these results may be applicable to more severe forms of IPV perpetration.” or similar.

6. Abstract: Conclusions: Please revise the first sentence to: Our results indicate that some mental disorders are associated with an increased risk of perpetrating IPV towards women, and that substance use disorders, as principal or comorbid diagnoses, have the highest relative and absolute risks.” or similar.

7. Abstract: Conclusions: Please revise the final sentence to: These findings support the idea that developing services for the assessment of IPV perpetration risk among men with substance use disorders could help to reduce the prevalence of IPV.” or similar.

8. Author Summary: Please use bullet points to denote separate items here.

9. Author Summary: “Why was this study done?”: Please remove “substantial” and “wide” from the first point, as they do not convey specific meaning.

10. Author Summary: Why was this study done?: The second and third points could be combined, for example: “Mental disorders are associated with increased risk of IPV perpetration; however, the nature and strength of the associations are uncertain because of limitations in study design and confounding factors in previous studies.” Or similar.

11. Author Summary: What did the researchers do and find?: We suggest combining the first and second points, for example: “We calculated the relative risk of IPV against women in men with psychiatric disorders identified in a Swedish population-based sample, and also compared the risk of perpetrating IPV in men with psychiatric disorders with their unaffected siblings.” or similar.

12. Author Summary: What did the researchers do and find?: For the second point, we suggest: “Most of the studied mental disorders were associated with a higher risk of IPV against women; the absolute rate of IPV against women ranged from 0.1% for autism and 2.1% for drug use disorders. Associated risks were two to seven times compared with the general population and two to four-fold compared with their unaffected siblings.” or similar.

13. Author Summary: What do these findings mean?: Please revise the first point to: “We found that several mental disorders are associated with increased risk of IPV against women, and the risk is further increased when there is a comorbidity with substance misuse.”

14. Introduction: 2nd paragraph: Please revise the second sentence to read: “Common deficits associated with mental disorders, such as impaired interpersonal skills and emotional dysregulation, and specific core symptoms of certain disorders, such as impulsivity manifested in individuals with ADHD and substance use disorders, and hostility exhibited in some people with mood disorders and antisocial personality disorder (9-12), have been linked to IPV against women (13, 14).” or similar. Also, please specifically reference your supporting literature for statements of “poor interpersonal skills”, “emotional dysregulation”, “impulsivity manifested in ADHD and substance use disorders”, and “hostility exhibited in some people with mood disorders and antisocial personality disorders”.

15. Introduction: 2nd paragraph: These two sentences seem to be saying essentially the same thing, please revise accordingly to consolidate: “Preliminary evidence suggests that individuals with mental illness have increased risk of IPV against women. A higher prevalence of IPV perpetration has been found in individuals with mental disorders than those without.” Please also provide references for this.

16. Introduction: 3rd paragraph: Please revise the abbreviation to “...Attention Deficit Hyperactivity Disorder (ADHD)...” Also, please spell out the definition of this term at its first instance, in paragraph 2 rather than in this paragraph.

17. Introduction: Final paragraph: Please revise the second sentence of this paragraph to: “To this end, we investigated the incidence of IPV in men with various mental disorders in a population-based sample using a longitudinal design.” or similar.

18. Methods: Outcome measure: Please revise the final sentence of this paragraph to: “The primary outcome was first IPV arrest after diagnosis.” or similar.

19. Results: Descriptives: Please show the absolute rates of IPV perpetrated by men towards women for the matched general population controls (as in Table 2 for the unaffected sibling data, or as a separate table) and refer to this in the text.

20. Results: Thank you for your response to Reviewer 1, Point R1.3. However, can you please also provide the complete set of hazard ratios for the confounders, (e.g. presented in a table).

21. Discussion: 2nd paragraph: Please revise the first sentence of this paragraph to: “Our findings underscore that risk of IPV in those with substance use disorders as the primary diagnosis had the highest risk among all studied disorders, and that substance use disorder comorbidity increased the risk of IPV for other disorders.” or similar.

22. Discussion: 3rd paragraph: Please provide a reference to support the final sentence: “This might be particularly helpful...focusing on the risk patients may pose in the context of drug and alcohol (which are more commonly used by individuals with mental disorders than the general population.”

23. Discussion: Page 17: Please revise the first sentence of this paragraph to: “Overall, we have shown that mental disorders, particularly substance use disorders, personality disorders, and ADHD, are associated with risk of IPV perpetration.” or similar.

24. Discussion: Page 18: Please revise the first sentence of this paragraph to: “Although our study is observational and causality cannot be inferred, if causality were to be assumed, then population risk percentages can be interpreted as…” or similar to avoid the implication of a causal relationship.

25. Discussion: Page 18: Please provide a reference for the statement: “It is likely that individuals with mental disorders selectively end up in abusive intimate partnerships, which could lead to reactive violence towards partners.”

26. Discussion: Conclusions: Please revise the final sentence to: “Our findings suggest that prioritizing the development of services to assess IPV perpetration among men with substance use disorders may help to reduce the risk of IPV against women.” or similar.

27. Figure 1: Please include in the legend that the bars represent the 95% confidence intervals, rather than indicating this on the y-axis label.

28. Table 3: There are no RHR values presented, although the title indicates that there are.

29. Table 3, Appendix 1, 3, 4 and 5: Please present unadjusted (crude) hazard ratios in addition to the adjusted results.

30. Appendix 5 Table: Please indicate in the legend that ratio of hazard ratios (RHRs) are presented.

Comments from Reviewers:

Reviewer #1: Thanks authors for their effort to improve the manuscript. I am satisfied with the response and the revision. No further issues needing attention.

Reviewer #3: In the context of my review of the original submission, the present version of the authors' manuscript is improved substantially. There is more careful attention paid to the narrative surrounding the theoretical factors that may account for the association between mental disorder and IPV, and the overall discussion is more in line with current theory and research in this area. Overall, this is a strong paper that is likely to make a significant impact on our understanding of IPV perpetration risk and in the development of novel prevention programs.

[LINK]

---

## [Editor Report · Decision Letter 2]

18 Nov 2019

Dear Dr. Fazel, 

On behalf of my colleagues and the academic editor, Dr. Phillipa Hay, I am delighted to inform you that your manuscript entitled "Mental disorders and intimate partner violence perpetrated by men towards women: a Swedish population-based longitudinal study" (PMEDICINE-D-19-02530R2) has been accepted for publication in PLOS Medicine. 

PRODUCTION PROCESS

PRESS

PROFILE INFORMATION

Thank you again for submitting the manuscript to PLOS Medicine. We look forward to publishing it. 

Best wishes, 

Caitlin Moyer, Ph.D.

Associate Editor 

PLOS Medicine

plosmedicine.org